# Nature’s Wind Turbines: The Measured Aerodynamic Efficiency of Spinning Seeds Approaches Theoretical Limits

**DOI:** 10.3390/biomimetics7040161

**Published:** 2022-10-12

**Authors:** Timothy C. A. Molteno

**Affiliations:** Department of Physics, University of Otago, P.O. Box 56, Dunedin 9016, New Zealand; tim@physics.otago.ac.nz

**Keywords:** Betz limit, aerodynamic efficiency, biomimicry

## Abstract

This paper describe a procedure to measure experimentally the power coefficient, $C_{p}$, of winged seeds, and apply this technique to seeds from the Norway maple (*Acer platanoides*). We measure $C_{p}=56.9\pm 2\%$ at a tip speed ratio of 3.21±0.06. Our results are in agreement with previously published CFD simulations that indicate that these seeds—operating in low-Reynolds number conditions—approach the Betz limit ($C_{p}=59.3\%$) the maximum possible efficiency for a wind turbine. In addition, this result is not consistent with the recent theoretical work of Okulov & Sørensen, which suggests that a single-bladed turbine with a tip-speed ratio of 3.2 can achieve a power efficiency of no more than 30%.

## 1. Introduction

Natural selection can lead to remarkable designs whose efficiency approaches fundamental limits. Biomimicry—using examples from the natural world to inspire solutions to engineering problems—has attracted interest in recent years as a means to design more efficient aerodynamic structures. For example, whale flukes [1,2] have inspired designs for minimizing turbulence in turbine blades [3] and bumblebees have inspired flapping-wing flyers [4].

Winged seeds of some trees use a novel technique to increase their dispersal—and thus gain a competitive advantage. Shaped like small wings, they begin to spin as they fall, and the lift from this spinning slows their fall. Designs that are more efficient extract more energy from the air flow fall more slowly and this increases the likelihood that they will land further from their parent tree, and thus increases their chances of surviving to adulthood.

Computational Fluid Dynamics (CFD) simulations have suggested [5,6] that these falling winged seeds (see Figure 1) have efficiencies that approach the Lanchester–Betz–Joukowsky limit [7,8,9]—often regarded as the maximum possible efficiency for a wind turbine.

The Lanchester–Betz–Joukowsky limit [10,11], hereafter “Betz limit”, states that no more than 59.3 % of the energy can be extracted from the wind by an un-shrouded wind turbine. This limit is considered optimistic, with more realistic models that take into account effects such as wake rotation leading to lower limits. For example, Glauert [12] showed the optimal wind turbine efficiency depends on the tip-speed-ratio, λ, the ratio of the speed of the tips of the turbine blades to the wind speed [10]. For typical tip-speed ratios seen in wind turbines, the Glauert limit is below the Betz limit by about 20%. The best human-engineered wind turbines have efficiencies approaching 50% [11]. Recent theoretical work of Okulov & Sorensen [13] further explores the theoretical limits to turbine efficiency, and concludes using vortex theory, that a single-bladed turbine with a tip-speed ratio of 3.2 can achieve a power efficiency of no more than 30%.

This paper described an experimental procedure to measure directly the efficiency of falling seeds as wind turbines. We then apply this technique to seeds of the Norway Maple (*Acer platanoides*), and find that our measurements are in broad agreement with previous numerical studies [6], and that the seeds have efficiencies that approach the Betz limit, and significantly exceed the theoretical limits derived by Okulov & Sorensen.

## 2. Theoretical Limits on Efficiency

Once the falling seed has settled into it’s spinning motion, it falls towards the ground with a constant velocity $v_{1}$. In this situation, the net upward aerodynamic force $F_{a}$, is balanced by the downward forces due to gravity.
(1)$$F_{a}=F_{g}\equiv mg,$$
where *m* is the mass of the seed, and *g* is the gravitational constant. Thus, measuring the seed mass establishes the magnitude of the aerodynamic force $F_{a}$. Using axial momentum-theory we can then derive an expression for $F_{a}$ in terms of some simplified parameters of the motion.

### 2.1. Axial-Momentum Theory

The simplest theoretical derivation of the Betz limit uses using a highly simplified model of flow called *axial momentum theory* [14]. This model assumes that air is incompressible, there is no heat transfer (dissipation), and the flow in and out of the rotating seed is axial and of uniform velocity. This aerodynamic force $F_{a}$ is produced by changing the momentum of the air as it flows past the seed. From the point of view of the seed, the air below the seed rushes upwards with a speed $v_{1}$, the air is slowed by the seed so that above the seed, the air is still moving upwards, but more slowly (with velocity $v_{2}<v_{1}$).

It can be shown (see for example [11,14]) that the velocity at the seed, *v*, is exactly half-way between $v_{1}$ and $v_{2}$. We use a factor *a* to describe how much the incoming speed, $v_{1}$, and outgoing speed, $v_{2}$, differ from the speed at the seed. The velocity before the seed is $v_{1}$, the velocity at the seed is $v_{1}\left(1-a\right)$ and the velocity after the seed is $v_{1}\left(1-2a\right)$, where *a* is the *axial induction factor*. This relationship between flow-tube area and flow speed shown in Figure 2.

The mass mass flow rate, $\frac{dm}{dt}$, through an area *A* is,
(2)$$\frac{dm}{dt}=\rho Av.$$
where *A* is the area, *v* is the velocity and ρ is the density. Conservation of mass requires that the mass flow rates through all of the flow areas, *A*, $A_{1}$ and $A_{2}$ are equal, i.e., $\frac{dm_{1}}{dt}=\frac{dm}{dt}=\frac{dm_{2}}{dt}$.

#### 2.1.1. The Force on the Seed

The aerodynamic force on the seed is the difference in the momentum of the air before the seed (per unit time), and after the seed. In other words,
(3)$$F_{a}=\frac{dmv}{dt}dt=\frac{dm_{1}}{dt}v_{1}-\frac{dm_{2}}{dt}v_{2}=\frac{dm}{dt}\left(v_{1}-v_{2}\right).$$
Substituting in for the mass flow rate (Equation (Equation 2)) through area *A*, and using the axial induction factor to find the velocity $v_{2}=v_{1}\left(1-2a\right)$ we get
(4)$$\begin{array}{ccc} F_{a} & = & \rho Avs.\left(v_{1}-v_{2}\right)=\rho Av_{1}\left(1-a\right)\left(v_{1}-v_{1}\left(1-2a\right)\right)\\ & = & \rho Av_{1}\left(1-a\right)v_{1}\left(1-1+2a\right)\\ & = & 2Aa\rho v_{1}^{2}\left(1-a\right). \end{array}$$

We can rearrange Equation (Equation 4) to get
(5)$$a\left(1-a\right)=\frac{F_{a}}{2A\rho v_{1}^{2}},$$
where we can measure everything on the right-hand side of this equation. Substituting the known force for $F_{a}$, (Equation (Equation 1)) we define a quantity *c* to be:(6)$$c\equiv \frac{F}{2A\rho v_{1}^{2}}=\frac{mg}{2A\rho v_{1}^{2}.}$$
Substituting *c* into Equation (Equation 5) gives a quadratic equation for *a*,
(7)$$a^{2}-a+c=0,$$
involving experimentally measurable quantities. Solving this we get an expression for *a*
(8)$$a=\frac{1\pm \sqrt{1-4c}}{2},$$
or
(9)$$a=\left[\frac{1}{2}-\frac{\sqrt{A\rho v_{1}^{2}-2F}}{2v_{1}\sqrt{A\rho }},\;\frac{1}{2}+\frac{\sqrt{A\rho v_{1}^{2}-2F}}{2v_{1}\sqrt{A\rho }}\right].$$
There are two solutions for *a*. However only one of these is physically possible—*a* must be less than 0.5, or the velocity after the seed is negative. This is
(10)$$a=\frac{1}{2}-\frac{\sqrt{A\rho v_{1}^{2}-2F}}{2v_{1}\sqrt{A\rho }}.$$

If *a* is known we can calculate the power as the product of the aerodynamic force (Equation (Equation 4)) and the velocity $v=v_{1}\left(1-a\right)$. The power becomes:(11)$$P=F_{a}\cdot vs.=2Aa\rho v_{1}^{3}{\left(a-1\right)}^{2}.$$

#### 2.1.2. The Power-Coefficient

The maximum possible energy that could be extracted from a wind turbine in an interval Δt, is limited by the kinetic energy $U_{k}$ of the air with mass $m_{1}$, flowing with velocity $v_{1}$ through the turbine area,
(12)$$U_{k}=\frac{1}{2}m_{1}v_{1}^{2}.$$
If the air flows through the turbine in Δt, the mass becomes $m_{1}=\frac{dm}{dt}\Delta t$, and where $\frac{dm}{dt}$ is the mass flow rate given in Equation (Equation 2). The kinetic energy is then
(13)$$U_{k}=\frac{1}{2}\frac{dm}{dt}\Delta tv_{1}^{2}=\Delta t\frac{A\rho }{2}v_{1}^{3},$$
and the maximum possible power $P_{\max}$, is
(14)$$P_{\max}=\frac{U_{k}}{\Delta t}=\frac{A\rho }{2}v_{1}^{3}.$$

The power coefficient $C_{p}$ is the ratio of the power to the maximum possible power. It is given by:(15)$$C_{p}\equiv \frac{P}{P_{\max}}=4a{\left(a-1\right)}^{2},$$
where *P* has been substituted from Equation (Equation 11). To find the maximum possible $C_{p}$, differentiate to get
(16)$$\frac{dC_{p}}{da}=4\left(a-1\right)\left(3a-1\right).$$
This has two solutions a=1, a minimum, and $a=\frac{1}{3}$ which maximises $C_{p}$ to be $\frac{16}{27}∼0.593$. This is the Lanchester–Betz–Joukowsky limit [15], the maximum possible power efficiency for a wind turbine.

### 2.2. Refined Limits Based on Vortex Theory

The Betz limit is derived assuming axial-momentum theory. Subsequent analyses have attempted to derive upper limits to efficiency based on more realistic models that take into account effects such as wake rotation.

Glauert [12] developed such a model—assuming an infinite number of blades, and showed that the optimum efficiency should be a function of the tip-speed ratio. The Glauert limit approaches the Betz limit for large tip-speed ratios λ>>10, and approaches zero as the tip-speed ratio approaches zero (see Figure 3). Recently [16] a closed-form expression for the $C_{P}$ under these assumptions was derived (this is plotted as the blue line in Figure 3):(17)$$\begin{array}{c} C_{P}=\frac{1}{\lambda^{2}}\left(a-1\right)\left(8a^{2}{\left(a-1\right)}^{2}\log\left(\frac{\lambda +\sqrt{-4a\left(a-1\right)+\lambda^{2}}}{2\sqrt{-a\left(a-1\right)}}\right)+\right.\\ \;\left.\lambda \left(\lambda^{3}+\sqrt{-4a\left(a-1\right)+\lambda^{2}}\left(2a\left(a-1\right)-\lambda^{2}\right)\right)\right). \end{array}$$

Several further limits have been suggested (see [13,17] for more details). Most indicate limits lower than the Betz limit; however, some imply that as the limit of the number of blades tends to infinity and the tip speed ratio is large, upper limits that exceed the Betz limit are provided: https://www.overleaf.com/project/6342675682757380f23d1c72, accessed on 15 September 2022.

## 3. Experimental Methods

Substituting for *a* from Equation (Equation 10) into the expression for $C_{p}$ (Equation (Equation 15)), yields an expression for $C_{p}$ in terms of measurable quantities.
(18)$$C_{p}=4a{\left(a-1\right)}^{2}=\frac{F}{A\rho v_{1}^{2}}+\frac{F}{A^{\frac{3}{2}}\rho^{\frac{3}{2}}v_{1}^{3}}\sqrt{A\rho v_{1}^{2}-2F}.$$

The only parameters required to determine $C_{p}$ are, *A*, ρ, $v_{1}$ and F=mg. This suggests an experimental procedure for calculating $C_{p}$. The equation for $C_{p}$ (Equation (Equation 18)) depends only on the following quantities:mass *m* of the seed,swept area, *A*, of the seed,falling terminal velocity $v_{1}$,density of air ρ.

The equipment needed to measure these is relatively straightforward—a diagram of a suitable setup involving two video cameras is shown in Figure 4. A balance is required to measure seed mass, and video capture is used measure the terminal velocity $v_{1}$, and the center of rotation.

### 3.1. Characterization of Experimental Uncertainty

When a quantity is not precisely known, for example, if the quantity is based on a measurement, then we will characterize the quantity as a probability distribution. The swept radius of the seed *r* is measured to be 38 mm with an uncertainty of 0.5 mm. In this case, we can choose to represent this quantity as U (37.5, 38.5)—a uniform probability distribution between 37.5 and 38.5. To compute the uncertainty in expressions that involve calculations of parameters with uncertain values, we draw samples from the distributions for each parameter and compute a distribution of the sampled expression. Statistics of this distribution—typically the 5th and 95th percentiles—are used to express the uncertainty in the resulting expression. For example, the swept area $A=\pi r^{2}$ can be calculated algebraically, or simulated by evaluating the function (in this case $\pi r^{2}$) drawing values for *r* from U (37.5, 38.5).

Figure 5 shows an example of such a histogram. This is the histogram of air-density values calculated from Equation (Equation 21) calculated using sampled values for air-temperature $T_{c}$ and humidity, as well as sampled assuming a Gaussian distribution of the $T_{c}$ and humidity. From this histogram, we can see that the percentiles are not very sensitive to the choice of uncertainty distribution, but that the range of possible values (tails of the distribution) are significantly wider when uncertainties are assumed to be Gaussian.

### 3.2. Falling Velocity Measurement

The horizontal camera films the vertical motion of the seed, and the time is measured (by counting frames) from the moment the seed intersects a laser level line (point *p* in Figure 4) until it touches the floor, (point *q* Figure 4). In the setup described here, this distance was 1.270 m ± 2 mm. To capture the uncertainty in this measurement, it is represented by a normal distribution with a mean of 1.27 m and a standard deviation of 2 mm (N (1.27, 0.002)). With a frame-rate of 100 FPS, this took 134 ± 1 frames (represented by a uniform distribution U (133, 135)). This resulted in a falling velocity, $v_{1}$ distribution of
(19)$$v_{1}=0.9478\pm 0.007.$$

### 3.3. Swept Area Measurement

The upward-looking camera is used to measure the axial rotation period of the spinning seed, as well as determining the center of rotation of the seed, and from this the area of the rotor. A still image showing the approximate center of rotation is shown in Figure 6. From this center of location, the span is measured as the distance to the rotor tip $r_{s}$, and the swept area $A=\pi r_{s}^{2}$

### 3.4. Air Density

The air density is calculated from the measured temperature, relative humidity and atmospheric pressure. This is done in two stages, first calculating the saturation pressure $P_{s}$ using Teten’s formula [18].
(20)$$P_{s}=6.1078e^{\frac{17.9T_{c}}{T_{c}+237.3}},$$
where $T_{c}$ is the temperature in Celsius. The partial pressure of water vapour $p_{v}$ is then given by $p_{v}=\phi P_{s}$ where ϕ is the measured relative humidity. The partial pressure of dry air is then $P_{d}=P_{a}-P_{v}$, where $P_{a}$ is the atmospheric pressure. The ideal gas equation, coupled with the molar mass of water vapour and dry air, gives the air density
(21)$$\rho =\frac{P_{d}M_{d}}{RT}+\frac{P_{v}M_{v}}{RT}$$
where R=8.314 is the universal gas constant, and *T* is the air temperature in Kelvin, and $M_{d}$ and $M_{v}$ are the molar masses of dry air and water vapour, respectively.

## 4. Results

The experimental procedure described in the previous section was applied to a seed rotor shown in Figure 1. Each measurement had an associated uncertainty, these are shown in Table 1. The mass *m* was measured using a Mettler digital balance as 0.19 g with an uncertainty of 0.01 g. Its measured value is represented as a uniform distribution between 0.18 and 0.20 g denoted by U (0.18, 0.20).

Using the measured values from Table 1, the falling speed calculation results in the best estimate of 0.9478 m·s${}^{-1}$, and explicitly incorporating uncertainty represented by 0.9478 ± 0.007. The swept area calculation is the largest contributor of uncertainty to the final result. The location of the center of rotation is imprecise (with an uncertainty of 1 mm), the swept radius is therefore modeled as U (37.5, 38.5). The swept area as calculated as $\pi r_{s}^{2}$ and this results in a swept area, $A∼0.004537\pm 0.0001m^{2}$.

By sampling from the distributions of parameters a histogram of $C_{p}$ values can be calculated, this is shown in Figure 7. The best estimate (median) of this distribution is 56.93%. The histogram also shows that 90 percent of this distribution lies between 54.6 and 59.3%. Alternatively using the mean and standard deviation of the distribution of samples for $C_{p}$, the experimental result can be stated as $C_{p}=56.9\pm 2.4\%$.

### Tip Speed Ratio

Glauert’s computations of optimum turbine efficiency taking wake rotation losses into consideration [13] indicate that the efficiency of an optimum turbine is lower than the Betz limit at low tip speed ratios. The measured tip speed, ωr, was 3.0426 ± 0.05. This gives a tip speed ratio of λ=3.21±0.03.

For the tip speed ratios in the region of 3.2, the Glauert-limit for optimum turbine efficiency is approximately 90% of the Betz limit, or 53.3% (values obtained from the closed-form expression in Equation (Equation 17)). The histogram of sampled results for $C_{p}$ shown in Figure 7 indicate that approximately 0.36% of simulated $C_{p}$ values lie below this Glauert limit. The histogram shows that it is highly likely (≥99%) that the efficiency of the seed rotor exceeds the Glauert limit.

## 5. Discussion and Conclusions

The analysis presented here is based on easily performed measurements on falling seeds. The results confirm that Norway-maple seeds are remarkably efficient with a power coefficient measured to be $C_{p}=0.569\pm 0.024$. This measurement is in agreement with CFD predictions [6]. Analysis of experimental uncertainties estimated from distributions of calculated $C_{p}$ values show that there is a ∼4.8% of chance that the measured $C_{p}$ value exceeds the Betz limit.

The measured tip-speed ratio is relatively low at 3.21±0.06. Glauert, taking wake rotation into account, derived a limit for turbine efficiency that for a tip speed ratio of 3.2 is 54.94%. Once again, analysis of the distribution of calculated $C_{p}$ values shows that it is 92% likely that the efficiency of a falling Norway-maple seed exceeds the Glauert limit.

Recent treatments have refined the Betz limit (see for example [13]) taking into consideration the finite number of blades as well as wake rotation. Okulov and Sorensen [13] find that with a tip-speed ratio of 3.2, the maximum a single-bladed turbine can only extract 30% of the available energy. The results we obtain ($C_{p}=0.569\pm 0.024$) far exceed this limit.

Modern large wind turbines achieve peak values for $C_{p}$ in the range of 0.45 to 0.50, [11] about 75% to 85% of the theoretically possible maximum. The falling seed achieves ∼95% of the Betz limit, a remarkable result, and in agreement with the CFD predictions of Holden et al. [6].

The upper limit on the amount of energy that can be extracted from air-flow by any device with projected area *A*, is usually given as the total kinetic energy contained in the wind that passes *through* the area *A* (see Equation (Equation 2)). This neglects the possibility that a wind turbine has an effect on the air outside its projected area. It is possible, at low Reynolds numbers, that the turbine’s influence on the flow is not confined to the projected area of the turbine. This would mean a larger area (and mass) of air was influenced by the seed, and therefore the expression for the maximum possible power, $P_{\max}$, in Equation (Equation 12) is artificially low. Since $P_{\max}$ appears on the denominator of the power coefficient $C_{p}\equiv \frac{P}{P_{\max}}$, underestimating this could allow $C_{p}$ values to exceed the theoretical limit.

Further research, perhaps visualizing the flow around spinning seeds could help determine if the true flow extent is significantly larger than the projected area of the spinning seed, and therefore provide a possible explanation for how these remarkable seeds can exceed theoretical limits on efficiency.

## Figures and Tables

**Figure 1 biomimetics-07-00161-f001:**
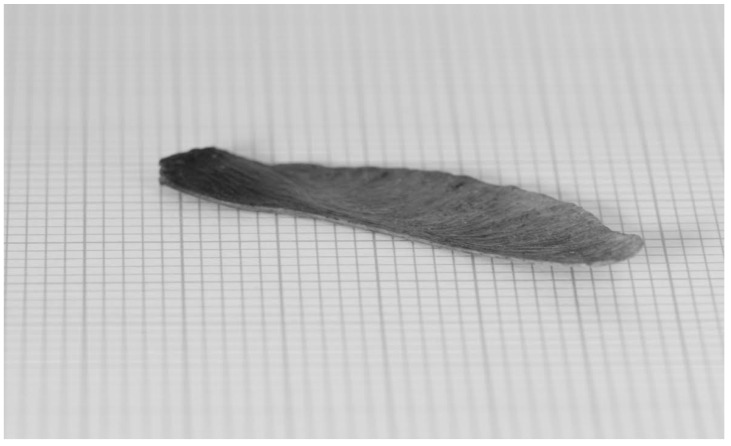
A seed from the Norway maple (*Acer platanoides*). The major grid spacing is 1cm. These seeds are wing-shaped, typically 3.5–5 cm long and with a chord of 1 cm. Their mass is between 100 and 200 milligrams when dry.

**Figure 2 biomimetics-07-00161-f002:**
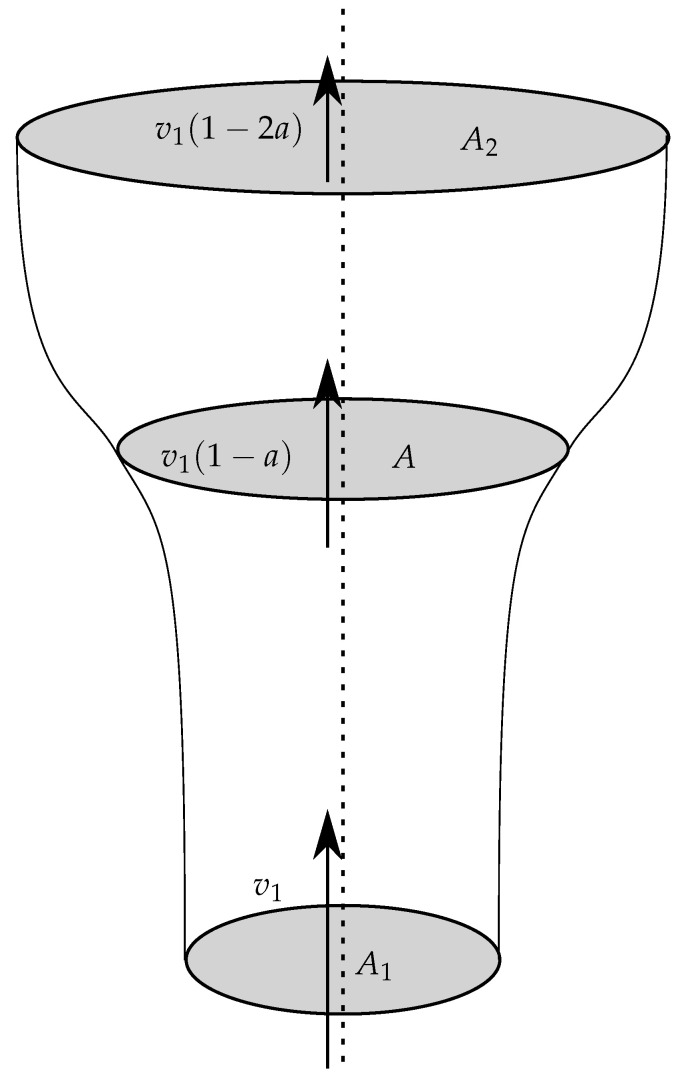
The flow tubes, before and after the falling seed. The wind approaches the disk of the seed with velocity $v_{1}$, and then slows down to velocity $v_{1}\left(1-2a\right)$ in the tube after the seed. At the seed, the velocity is half-way between $v_{1}\left(1-a\right)$. The areas of the flow tube are $A_{1}$, *A* and $A_{2}$ before, at and after the turbine, respectively.

**Figure 3 biomimetics-07-00161-f003:**
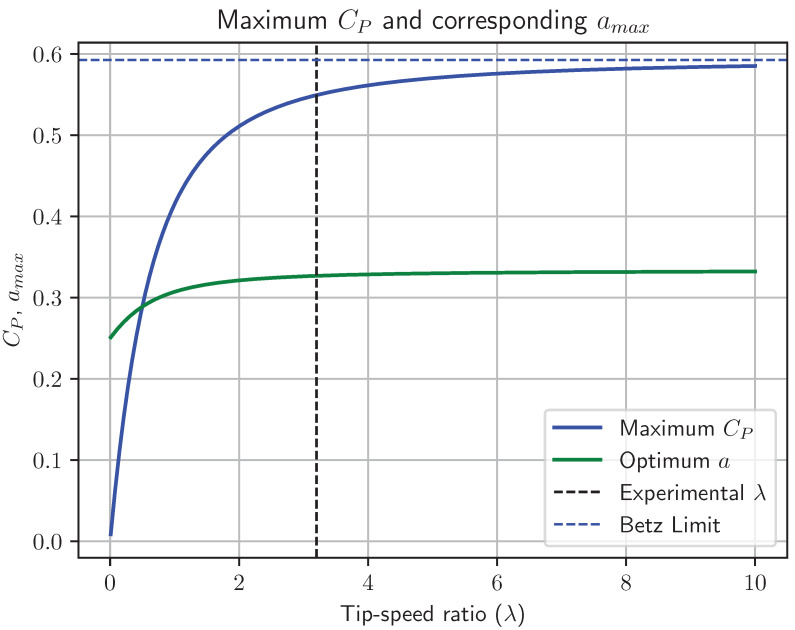
Glauert limits for an optimum wind maximum turbine as a function of tip-speed-ratio. The axial induction factor that maximises $C_{p}$ is shown as $a_{max}$, as well as the maximum $C_{p}$. At large tip-speeds, $C_{p}$ converges to the Betz limit of 0.593, and the optimum axial induction factor converges to $a_{m}ax=\frac{1}{3}$. The vertical line indicates the tip-speed ratio for the falling seed studied here.

**Figure 4 biomimetics-07-00161-f004:**
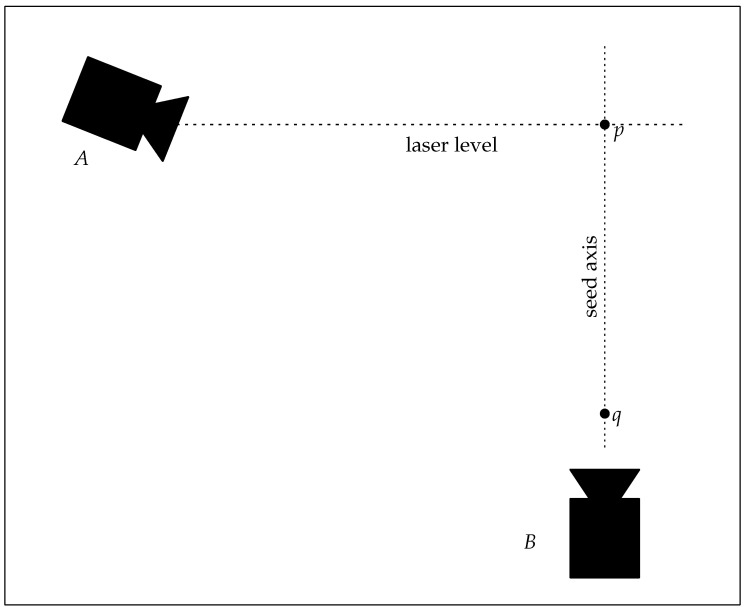
Diagram of apparatus for measuring seed turbine efficiency. Camera *A* looking horizontally, is used to determine the terminal falling speed, while camera *B* looking upwards, is used to measure the center of rotation, and hence the swept area *A*.

**Figure 5 biomimetics-07-00161-f005:**
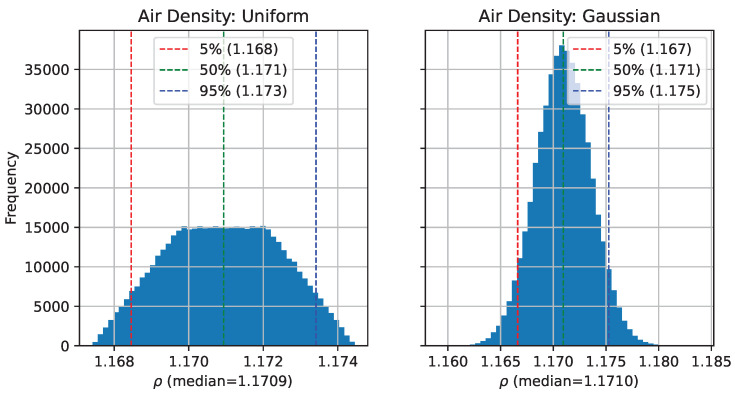
Histogram of 500,000 samples of air density values calculated with different uncertainty distributions for temperature $T_{c}$ and humidity (see Equation (Equation 21)). In the left plot, the uncertainties are uniform distributions, and in the right plot, they are Gaussian distributions. The median, and 5–95% ranges are not significantly changed by these different choices; however, the tails of the distribution are significantly wider when uncertainties are assumed to be Gaussian.

**Figure 6 biomimetics-07-00161-f006:**
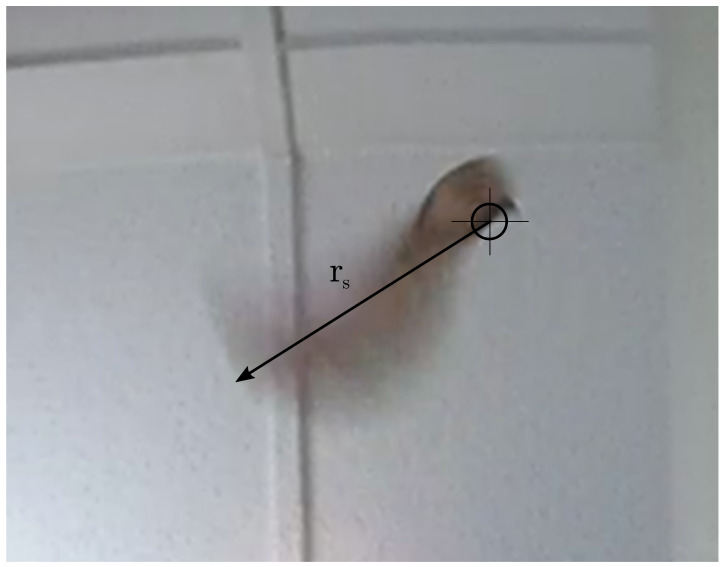
Still image from the upward-looking camera used to estimate the center of rotation. The approximate center of rotation is shown by the cross-hairs. The center of rotation is estimated by inspection using the differential blurring caused by the rotation. The spinning radius $r_{s}$ is then estimated from this point to the top of the motion-blurred seed.

**Figure 7 biomimetics-07-00161-f007:**
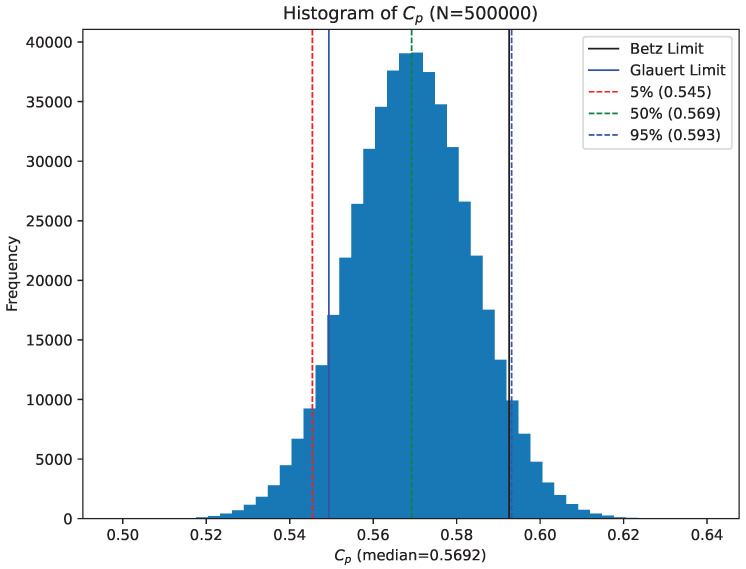
Histogram of 500,000 randomly generated $C_{p}$ values for a single seed, showing the experimental uncertainty for the estimate of $C_{p}$. The Betz limit is shown as the black dotted line close to the 95th percentiles of the $C_{p}$ distribution. For this seed, 95% of $C_{p}$ values lie in the range 0.569 ± 0.023.

**Table 1 biomimetics-07-00161-t001:** Experimental parameters measured from a falling Norway maple seed. Uncertainties are expressed as probability distributions, with *U* denoting a uniform distribution, and the ± symbol denoting a Gaussian distribution where the uncertainties are one standard deviation.

Quantity	Units	Symbol	Value with Uncertainty
mass	g	*m*	U (0.18, 0.20)
Rotation Frequency	Hz	*f*	12.744 ± 0.12
Falling speed	m·s${}^{-1}$	$v_{1}$	0.9478 ± 0.007
Tip Speed	m·s${}^{-1}$		3.0429 ± 0.05
Tip Speed Ratio		λ	3.211 ± 0.033
Reynolds Number		Re	2042.6 ± 22.5
Swept radius	mm	$r_{s}$	U (37.5, 38.5)
Swept area	m${}^{2}$	*A*	0.004537 ± 0.0001
Air Temperature	C	*T*	N (23.9, 0.3)
Relative Humidity	%	ϕ	N (68.6, 1.0)
Air Density	kg·m${}^{-3}$	ρ	1.17092 ± 0.004

## Data Availability

Not applicable.

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
