# Peer review of "Nature’s Wind Turbines: The Measured Aerodynamic Efficiency of Spinning Seeds Approaches Theoretical Limits"

_biomimetics, 2022, doi:10.3390/biomimetics7040161_

Round 1
Reviewer 1 Report
This paper introduces an experimental method to measure the power coefficient Cp of the seed wings of Acer norwegiana, and applies this method to the seed of Acer norwegiana. The mathematical modeling and experimental research are carried out. The overall structure of the paper is rigorous, and the language is appropriate. There are some small points to be noted:
(1) Formula (3) Fa=2Aa ρ How is v21 (1 − a) derived? Please give a detailed description.
(2) Is it possible to reach 1 when Cp value in Figure 3 reaches 0.6?
(3) The image in Figure 6 is very unclear. It is better to replace it.
(4) In the conclusion of the paper, it is mentioned that the power extracted by the turbine from the air may exceed the maximum kinetic energy of the wind passing through the projection area, that is, Cp exceeds 1.0. It is better to provide specific experimental data or graphics. This experimental conclusion should be compared with the theoretical results? The purpose is to explain this phenomenon more accurately.
Author Response
I would like to thank the reviewer for their review. I had addressed their comments as follows
- I have added more detail to the derivation of the aerodynamic force as requested.
- I have added a note to the caption of Figure 3, showing that the limiting value for C_p is 0.593 in the limit of large tip-speeds.
- I have added a clarification in the conclusion on how underestimating the effective area of a turbine could cause an over-estimate of the power-coefficient.
Reviewer 2 Report
Notes in the attachment.

Author Response
I would like to thank the reviewer for their review. I have addressed the highlighted comments. I have numbered all equations and added punctuation as requested, and where larger changes have been made, I address them below (broken up by page number of the original manuscript)
Page 2:
- Added some characteristic parameters (length, chord and mass) of the Norway maple seeds studied here.
- Corrected the axial ratio in the table of symbols
- Changed the expression \dot{m} for the mass flow rate to \frac{d m}{d t} to make it more readable.
Page 3:
- Added punctuation to equations (on all pages)
- Added explanations of parameters $A$ $\rho$ and $v$ and added to the symbol table
Page 4:
- Numbered all equations as requested.
- Added an explicit statement on how the power is calculated.
Page 5:
- Added a note to the caption explaining the meaning of the experimental \lambda.
- Added the tip speed ratio to the table of symbols and included a reference to where it is defined it in the Introduction section.
- Plotted the line on the graph for the explicit Glauert limit as requested
Page 6:
- Added T_c, R, M_d and M_v to the table of symbols
- Elaborated on how uncertainties are calculated using histograms
- Added an explanation of M_d and M_v to the text
Page 7:
- I have elaborated on what uncertainties are shown in the figure.
- This was mistakenly referred to as area in the text when it is actually air density (this has been corrected)
Page 8:
- I have added a better estimate of the centre of rotation as requested.
- Updated to explain how the center of rotation is estimated.
- The radius of the swept area is added to the diagram
- $r_s$ is added to the list of symbols
Page 9:
- Updated table description and column names as suggested.
- Indicated explicity in the table caption how uncertainties are expressed (either as a uniform distribution or a gaussian)
- I have updated the tip speed ratio expression in the text and elaborated on how it is calculated.
- Angular velocity is included in the table of symbols.
- I have made it clear that the estimates of uncertainties are made from the statistics of samples drawn from calculated values
Page 10:
- I have reworded that last sentence to make it less categorical
- I have added a suggestion that further research be done to explore whether the true flow extent exceeds the projected area significantly enough to explain how these remarkable seeds can show efficiencies that exceed the Glauert-limit.